# Important Hormones Regulating Lipid Metabolism

**DOI:** 10.3390/molecules27207052

**Published:** 2022-10-19

**Authors:** Dengke Zhang, Yanghui Wei, Qingnan Huang, Yong Chen, Kai Zeng, Weiqin Yang, Juan Chen, Jiawei Chen

**Affiliations:** 1Department of Surgery, The Eighth Affiliated Hospital, Sun Yat-sen University, Guangzhou 510275, China; 2School of Biomedical Sciences, The Chinese University of Hong Kong, Hong Kong 999077, China; 3Department of Medicine & Rehabilitation, Tung Wah Eastern Hospital, Hong Kong 200051, China

**Keywords:** lipid metabolism, hormones, regulation network

## Abstract

There is a wide variety of kinds of lipids, and complex structures which determine the diversity and complexity of their functions. With the basic characteristic of water insolubility, lipid molecules are independent of the genetic information composed by genes to proteins, which determine the particularity of lipids in the human body, with water as the basic environment and genes to proteins as the genetic system. In this review, we have summarized the current landscape on hormone regulation of lipid metabolism. After the well-studied PI3K-AKT pathway, insulin affects fat synthesis by controlling the activity and production of various transcription factors. New mechanisms of thyroid hormone regulation are discussed, receptor α and β may mediate different procedures, the effect of thyroid hormone on mitochondria provides a new insight for hormones regulating lipid metabolism. Physiological concentration of adrenaline induces the expression of extrapituitary prolactin in adipose tissue macrophages, which promotes fat weight loss. Manipulation of hormonal action has the potential to offer a new therapeutic horizon for the global burden of obesity and its associated complications such as morbidity and mortality.

## 1. Introduction

Lipids are the general name of fats and lipoids. Fats are triglycerides (TG); lipoids include cholesterol and its esters, phospholipids, glycolipids. Triglycerides play an important role in energy storage and supply. Exogenous triglycerides are mainly obtained from food, while endogenous triglycerides are mainly synthesized by the liver, adipose tissue, and small intestine. In the liver, they are assembled with apolipoproteins, phospholipids and cholesterols to form very low-density lipoprotein (VLDL) and then is transported out of the liver. Disorder of VLDL synthesis will lead to the accumulation of triglycerides in liver cells—namely fatty liver. Because the basic substrates for fatty acid synthesis, such as acetyl coenzyme A, ATP and NADPH are mainly derived from glucose catabolites, triglyceride metabolism is associated with glucose metabolism. In addition to the metabolic changes in the synthesis decomposition balance caused by changes in substrates supply, hormones also affect fat metabolism. Hormones like adrenaline, norepinephrine, and glucagon triggered by fasting, starvation and sympathetic excitement stimulate fat mobilization. Cyclization of adenylate further activates cAMP-dependent protein kinase and increases adipose triglyceride lipase (ATGL) activity. However, the key enzyme for fatty acid synthesis, Acetyl-CoA carboxylase, is inhibited. In turn, insulin plays an anti-lipolysis role, which activates acetyl coenzyme A carboxylase to promote fatty acid synthesis, and insulin increase fat storage in adipose tissue. As an important component of TG, cholesterol esters and phospholipids, fatty acid can be used to synthesize strong bioactive substances such as prostaglandin, leukotrienes, and thromboxane A2, who play a role in regulating local immunity. Phospholipids and their derived components are the basic components of biological membranes. In the inner layer of cell membranes, phosphatidylinositol-4,5-diphosphate plays a role in transmitting cell signals. Cholesterol is also an important component of bio-membrane. In some endocrine glands, cholesterol is used to synthesize steroid hormones. However, high concentration of cholesterol in the blood can induce atherosclerosis. For cholesterol synthesis, HMG-CoA reductase is a crucial enzyme. Insulin and thyroid hormones induce the synthesis of HMG-CoA reductase to promote the process, while thyroid hormones promote the conversion of cholesterol into bile acid. Glucagon chemically modifies and regulates the phosphorylation of HMG-CoA reductase and inactivates it, while cortisol directly inhibits its activity. In the circulation, blood lipids are mainly transported and metabolized in the form of plasma lipoproteins. Plasma lipoproteins are divided into several main types of lipid and protein contents, with different physiological functions. As mentioned above, VLDL is the main form for transporting endogenous TG, while low-density lipoprotein (LDL) mainly transports endogenous cholesterol. Oxidative modified LDL and VLDL are considered as important pathogenic factors of cardiovascular and cerebrovascular diseases. High-density lipoprotein (HDL) is considered to be a protective factor of cardiovascular and cerebrovascular diseases, due to its ability to transport the peripheral cholesterol to the liver for metabolism.

## 2. Insulin and Transcriptional Regulation

Insulin plays a central role in lipid metabolism—promoting energy storage and inhibiting energy release; this can be shown especially in insulin-resistant individuals who will develop fat accumulation in the liver, a condition induced by increased de novo lipogenesis [1]. Accumulating studies have shown phosphoinositide 3-kinase (PI3K)-Akt pathway to be a signaling cascade; being integral to the metabolic actions of insulin, after insulin binding to its cognate receptor, PI3K is recruited to phosphorylate insulin receptor substrates (IRS) and generates 3′phosphoinositides. Phosphatidylinositol (3,4,5)-trisphosphate (PIP3) generation promotes the recruitment of pyruvate dehydrogenase kinase 1 (PDK1) and Akt (also known as protein kinase B), leading to the subsequent phosphorylation of Akt by PDK1. This phosphorylation can be also induced by mammalian targeting of rapamycin (mTORC) 2, but within a different amino acid site [2]. Then, multiple downstream pathways such as mTORC1, glycogen synthase kinase and the FoxO transcription factors were signaled to control glucose and lipid metabolism [3,4].

As the function of AKT in human energy and material metabolism is elucidated, more downstream pathways of Akt are being studied extensively. This includes the sterol regulatory element-binding protein 1c (SREBP1c) and the carbohydrate response element-binding-protein (ChREBP), which belong to lipogenic transcription factors activating lipogenic genes such as Fasn, ACC, Scd1, and Elovl6 [5]. During the synthesis of fatty acids, TGs, cholesterol and its esters, SREBPs are a class of transcription factors that exist widely and play an integral role [6]. Insulin activation of Akt enhances both the synthesis and processing of SREBP1c, which is the predominant SREBP subtype in hepatocytes; in SREBP1c precursor processing, insulin enhances the affinity of the SREBP1c precursor complex for vesicles and Golgi apparatus, and increases proteolytic cleavage of the precursor protein induced by SREBP cleavage-activating protein (SCAP) in an PI3K-AKT-dependent way [7]. The requirement to SREBP1c in some pathological states, including insulin resistance-induced hepatic fat accumulation, has been proved in SCAP-specific knockout mice models [8]. SREBP1c also serves as a connection between lipid and glucose metabolism. Under low ATP supply status, the upregulated AMP-activated protein kinase (AMPK) is present in hepatocytes phosphorylates SREBP1c and thus inhibits de novo lipogenesis (DNL). The activation of AMPK could protect high-fat feeding rodents from hepatocytes steatosis or atherosclerosis [9]. Furthermore, SREBP1c has been shown to play an important role in adipocyte differentiation [10]. ChREBP is a glucose-responsive transcriptional factor. Specific inhibition of ChREBP in mice hepatocytes reduces DNL and TGs [11]. However, it is still elusive whether SREPB1c and ChREBP are activated in insulin-resistant individuals, as increased fat synthesis and storage could be observed in these patients [12]. More extensive studies are needed on the interaction between these transcription factors and their role in the human body.

Similar to SREBPs and ChREBP, FoxO proteins are also transcriptional factors controlled by Akt through a delicate phosphorylation action, which mainly inhibit the expression of lipogenic target genes [13]. The transcriptional mechanism of FoxO1-controlling liver lipid metabolism has not been fully determined. For the SREBP1c promoter, there is evidence which shows that FoxO1 binds to it directly and reduces its transcriptional activity [14]. The glucokinase, as a lipogenic gene that functions by increasing the glucose-6-phosphate level and activating the subsequent ChREBP, is also affected by FoxO1. Based on the current evidence, AKT-dependent inhibition of FoxO1 seems to be required for DNL [15]. Experiments have yielded conflicting results still on the role FoxO1 plays in triglyceride secretion. It was reported that overexpression of FoxO1 in transgenic mouse hepatocytes resulted in elevated ApoCIII and subsequent hypertriglyceridemia [16]. Similarly, FoxO1 was reported to be necessary for the expression of microsomal triglyceride transfer protein (MTP), which led to the secretion of TG-rich VLDL [17]. However, Zhang et al. [18] reported decreased serum TG levels due to the failure to repeat Foxo1-reduced VLDL triglyceride content and ApoCIII-induced serum TG rising in vivo or in vitro. Consistent with this finding, hepatic specific deletion of FoxO1 does not alter serum TG levels. Insulin signaling leads to Foxa2 inactivation through Thr156 phosphorylation and nuclear exclusion, which inhibits Foxa2 target gene expression and decreases hepatic lipid metabolism [19]. Characterization of FoxO6 as an integrator of hepatic insulin signaling with TG-rich VLDL (VLDL-TG) production in the liver provides important insights into the mechanism of hypertriglyceridemia. FoxO6 did not undergo insulin-dependent phosphorylation and nuclear translocation like other FoxO subtypes did [20]. FoxO6 promoted lipogenesis and augmented VLDL-TG secretion, which was demonstrated by Kim and his colleagues [21]. This action correlated with the ability of FoxO6 to stimulate hepatic production of MTP, which is a key to enzyme-promoting VLDL assembly and secretion by catalyzing lipid transferring to nascent apolipoprotein B (apoB) [22]. Although controversies remain on this topic, there is no doubt that FoxO, especially FoxO1, plays an important role in insulin-mediated lipid metabolism. Insulin signaling pathway is summarized as Figure 1.

A growing amount of evidence from in vivo studies supported the idea that MTORC1 plays a role in activation of the AKT downstream pathway [23]. Despite the complexity of Akt downstream pathway, the most critical point is the regulation of lipid metabolism and the progression of some abnormal physiological status which requires insulin signaling via Akt.

## 3. Glucagon

The function of glucagon is in promoting energy release and inhibiting energy storage in response to low circulating glucose levels, which are largely controlled by insulin [24]. Besides its role in blood–glucose regulation, patients treated with glucagon receptor antagonists (GRA) often suffer from dyslipidemia, fatty liver and weight gain, suggesting that glucagon play an important role in lipid metabolism [25]. Glucagon acts mainly on the hepatocytes which possess the highest level of glucagon receptors. Following mechanisms may be involved during this process: (1) At the molecular level, following glucagon binding to its receptor on the hepatocyte, cAMP will be activated and accumulated, which in turn activates cAMP-response element-binding protein (CREB). As a result, the transcription of carnitine acyl transferase (CPT-1) increases, which converts fatty acids into acylcarnitine and activates β-oxidation-increasing fatty acid catabolism [26]. In addition, glucagon binds to the receptor and induces PKA-dependent phosphorylation, leading to inactivation of acetyl-CoA carboxylase, which is a key enzyme in malonyl-CoA synthesis. Since malonyl-CoAs suppresses β-oxidation by inhibiting CPT-1 activity, downregulation of malonyl-CoA will lead to more free fatty acids (FFAs) entering mitochondria for β-oxidation, rather than being released into circulation from hepatocytes in the form of VLDL after re-esterification [27]. Under physiological conditions, glucagon is sufficient to activate fatty acid oxidation gene expression; on the contrary, the insulin-PI3K-AKT pathway meditates the inhibition of Foxa2 through the Thr156 site phosphorylation and nuclear exclusion mechanism [19]. Glucagon signaling-activated adenylate cyclase (AC), on the one hand, elevated extracellular cAMP levels, augmented the activity of peroxisome proliferator-activated receptor α (PPAR α) by direct phosphorylates AMP-activated protein kinase (AMPK), which increased the transcription of fatty acid oxidation genes such as Aox and Cpt1a. On the other hand, activated AMPK inhibited the transcriptional activity of SREBP1c and ChREBP, leading to decreased lipid synthesis [28,29]. In conclusion, glucagon reduces de novo fatty acid synthesis and thereby reduces VLDL release. Besides, it is reasonable that glucagon signaling may increase the AMP/ATP ratio, which is required for AMP-activated kinases activation. This will induce transcription of β-oxidation-related genes by enhancing the expression of PPAR α [30]. There were in vivo experiments suggesting the importance of glucagon in hepatic lipid metabolism. In rats and mice, chronic physiological increased glucagon concentrations in plasma, increased mitochondrial oxidation of fat in liver and reversed diet-induced hepatic steatosis. However, these effects were abrogated in inositol triphosphate receptor 1(INSP3R1)-knockout mice, suggesting that the mechanisms by which glucagon affects are also mediated by stimulation of the INSP3R1 [31]. For obese mice, cAMP injection increased extracellular cAMP level and ameliorated the impaired lipid metabolism. This further confirmed the important role of cAMP-mediated PPAR α activation through AMPK in glucagon regulation of lipid metabolism. When mice were injected with glucagon, decreased plasma FFA, TG concentrations, hepatic TG synthesis and secretion were observed [32].

The lipolysis of lipid droplets in adipocytes depends on PKA-dependent HSL phosphorylation and perilipins on its surface [33], resulting in FFAs and glycerol release into circulation eventually. Discovery of glucagon receptor mRNA in rat adipocytes [34] supported the effect of glucagon on HSL [34] and subsequently lipolysis [35]. The effect of glucagon on HSL remained controversial, especially in humans; there was no evidence of the existence of glucagon receptor expression on human adipocytes. According to the reported literature, glucagon-induced lipolysis can be observed under the action of glucagon with hyper-physiological concentration; however, this lipolysis can be eliminated by insulin in some other studies [36,37,38], which is consistent with the powerful anti-lipolytic effects of insulin. Therefore, if there is any such lipolysis of human adipocytes caused by glucagon, it is physiologically significant under the premise of low insulin secretion [39,40,41].

## 4. Thyroid Hormone

As critical regulators of metabolism, development and growth, thyroid hormones have prominent effects on both cholesterol and fatty acid metabolism. Hypothyroidism leads to elevated levels of LDL cholesterol, HDL cholesterol and TGs in the serum, while hyperthyroidism does the opposite [42,43].

(1)The role of thyroid hormone receptors in lipid metabolism.

Thyroid hormones regulate a large panel of genes related to lipogenesis by binding to lipid’s specific receptor, which is a ligand-dependent transcription factor [44,45]. There are two major isoforms of receptors, α (THRα) which is mainly found in the heart and bone, while β (THRβ) is highly expressed in the liver [46,47]. In the absence of ligands, THRs bind to thyroid hormone response elements (TREs) and represses the transcription of target genes by recruiting a co-repressor complex. This will be released due to conformational changes in THRs in the presence of ligands, typically accompanying recruitment of a co-activator complex to activate target genes [48,49]. The effect of THRs and subsequent pathways in lipid metabolism is complex and still under investigation. THRβ mutation mice with a dominant negative effect on THRβ genes will develop hepatocyte steatosis, which indicates the role of THRβ; this phenomenon is attributed to decreased THR-mediated fatty acid β-oxidation and increased PPARγ signaling [50]. Additionally, consistent with this, THRβ targeting agonists to the liver reduces hepatic triglyceride content [51]. By contrast, THRα gene locus mutation and THRα-loss mice with dominant negative effect on THRα genes led to decreased lipogenesis and weight loss, suggesting THRα elevates lipid synthesis and accumulation [50,52]. However, male mice will exhibit hepatic steatosis after being introduced to a dominant negative Pro398His mutation into the Thra gene locus, owing to Pro398His mutant receptor interfering with the binding of PPARα, which mediates transcription of fatty acid oxidation genes to its promoter response element [53]. This suggests that under normal circumstances, PPARα is also downstream of THRα and receives its signal, promoting lipid oxidation. As we can see, the precise mechanism that underlies the differences in hepatic lipid metabolism between those with various mutations and knockout mice is still unknown yet, which warrants further study.

In addition to regulating the expression of lipogenic genes directly, TH also affects the activities of other transcription factors such as SREBP1c and ChREBP to affect liver lipogenesis indirectly [54]. TH increases ChREBP expression in hepatocytes by recruiting THRs to ChREBP promoters [55]. How thyroid hormones regulate SREBP1c in humans remains elusive. In mice, thyroid hormones regulated SREBP1c transcription negatively through a putative negative thyroid hormone response element (nTRE) [56]. However, another study suggested that SREBP1c transcription is also upregulated by non-genomic thyroid hormone signaling [57]. In addition, thyroid hormones can affect lipid metabolism by increasing FoxO1 nuclear localization, DNA binding and target gene transcription in a THRβ-dependent manner [58]. Although thyroid hormone increases the expression of genes involved in DNLs, it does not cause a net increase level of triacylglycerol in mouse hepatocytes [59], which mainly due to upregulated metabolism of FFAs by thyroid hormones.

(2)TH increases hepatic TG synthesis.

Fatty acids are basic substrates for liver TG synthesis. Their major source is circulating free fatty acids (FFAs), which originate from the lipolysis of white adipose tissue and dietary fat sources. FFAs enter hepatocytes in a protein transporter-dependent manner. TH not only promotes the lipolysis procedure, increasing circulation FFAs concentrations, but also induces the protein transporter expression such as fatty acid transporter proteins (FATPs), liver fatty acid-binding proteins (L-FABPs) and fatty acid translocase [60,61,62]. A study in 2009 suggested fatty acid uptake from triglyceride-rich lipoproteins was influenced by thyroid hormones in a tissue-specific manner, as hyperthyroidism increased triglyceride-derived fatty acid uptake in liver and muscle, whereas hypothyroidism increased triglyceride-derived fatty acid uptake in white adipose tissue and decreased its uptake in liver [63]. The exact mechanism by which thyroid hormones alter FFA uptake in the liver has not been fully elucidated. In addition to the exogenous FFAs in circulation, another source of FFAs for TG production was glucose, which was more significant under excess dietary intake. This endogenous process of FFA generation from glucose was called de novo lipogenesis (DNL). DNL is an enzymatic cascade process, which depends on the catalysis of a variety of key enzymes regulated by nutritional status and hormones [64]. Thyroid hormones are well-known inducers of hepatic DNL for stimulating the transcription of several key genes involved in lipogenesis in rodents, such as acetyl-CoA carboxylase alpha (Acc1; also known as Acaca), fatty acid synthase (Fasn) [65]. FFAs are esterified to triacylglycerol, after which they can be packaged into VLDL and stored as fat droplets. Thyroid hormones reduce apolipoprotein B100 (Apo B100) levels in the livers of rats, leading to decreased production of VLDL and LDL [66].

(3)TH reduces the content of TG in hepatocytes and adipocytes by promoting TG catabolism.

During hyperthyroidism, there is a net reduction in total hepatic triglycerides [67] due to more active catabolic actions of hepatic lipids, although thyroid hormones stimulate lipogenesis. The main role of thyroid hormone on hepatic lipids catabolism is the mobilization of FFAs from stored triacylglycerol and following β-oxidation. Hepatic lipase and adipose triglyceride lipase (ATGL) are two key enzymes catalyzing the stored TG decomposition in hepatocytes [68]. The expression and activity of hepatic lipase are related to thyroid hormone levels [69]. Studies of subclinical hypothyroidism have shown that reduced thyroid function is associated with decreased hepatic lipase activity, which can be restored by exogenous thyroid hormone replacement treatment [70,71]. The effect of thyroid hormones on ATGL expression and activity in hepatocytes is still unknown. A study in 2015 about cultured rat hepatocytes suggested that the ATGL recruitment to the surface of lipid droplet (LD) reduced lipid storage and the activity of LD-related proteins, which was related to TH level [71]. It was reported that thyroid hormones increase the expression of zinc-α2-glycoprotein in hepatic cells [72], which stimulated lipolysis in humans and reduced body fat in mice [73]. In addition, TSH has an inhibitory effect on ATGL expression in mature adipocytes, which is assumed to be related to PKA activation [74]. As serum TSH concentration changes with the feedback of TH levels, interaction between thyroid hormones and TSH might also relate to the lipolytic action of thyroid hormones. A report in 2015 suggested that TSH inhibits hepatic cholesterol synthesis by reducing HMG-CoA reductase phosphorylation through AMPK [75]. TSH also suppresses the synthesis of hepatic bile acid via an SREBP2-hepatocyte nuclear factor 4α (HNF4α)–CYP7A1 signaling pathway [76]. However, the TSH rise accompanies hypothyroidism with decreasing serum TH concentrations. It is difficult to confirm whether the effect of TSH on lipid metabolism is independent of TH. Lysosomal acid lipase (LAL) is another critical factor regulating hepatic triacylglycerol lipolysis [77]. Lipophagy is a specific type of autophagy which involves the engulfment of triacylglycerol stored in the fat droplets by autophagosomes, followed by autophagosomal–lysosomal fusion and eventually triacylglycerol degradation and hydrolysis into FFAs [78,79]. Thyroid hormone increases the number of lipid-laden autophagosomes and lysosomes in hepatic cells in a THR-dependent manner [80].

(4)Mitochondria, an important target for TH regulating lipid metabolism.

As the main sites of fatty acid metabolism, mitochondria are also the main target of thyroid hormone regulating lipid metabolism in the liver [81]. The biogenesis and function of mitochondria mainly depend on the regulation of nuclear, but the possession of circular DNA genetic material gives it a semiautonomous position. TH affects lipid metabolism by coordinating signals originating from two parts [82]. Thyroid hormones upregulate the level of PPARγ co-activator 1α (PGC1α), which acts as a co-transcriptional factor by activating nuclear respiratory factor 1 (NRF1) that promotes the expression of mitochondrial transcription factor A (mtTFA) and consequently induces mitochondrial biogenesis [82]. This axis is the basic mechanism of nuclear regulation of mitochondrial content by thyroid hormones [82]. In addition to this axis, it was reported that THRs also existed in mitochondria, regulating the transcription of mitochondrial genome [83]. For mitochondrial FFAs β-oxidation in hepatocytes, carnitine O-palmitoyl transferase 1 liver isoform (CPT1-Lα) is the key enzyme whose transcriptional activity was stimulated by TH [84]. Activation of SIRT1 activity, mediated by thyroid hormones, was proved to induce PGC1α activity and increase CPT1A mRNA expression [85]. Thyroid hormones also regulate CPT1A gene expression by increasing PPARα signaling in the liver [49]. Thyroid hormones increase the expression of other mitochondrial enzymes crucial for fatty acid β-oxidation, such as medium-chain acyl-CoA dehydrogenase (MCAD) [86], pyruvate dehydrogenase kinase isoform 4 (PDK4) [87] and mitochondrial uncoupling protein 2 (UCP2) [88].

(5)TH reduces serum cholesterol levels.

TH has a profound impact on basic serum cholesterol levels maintenance and the whole cholesterol metabolism process in vivo, including hepatic synthesis, export of cholesterol in the form of LDL, reverse transporting from peripheral tissues, hepatic reuptake mediated by LDL receptors (LDLRs) and excretion into the intestine after conversion into bile acids [89]. In the hepatocytes of rats, thyroid hormones induce the expression of hydroxymethylglutaryl-CoA reductase (Hmgcr) and farnesyl pyrophosphate synthetase (Fdps) to promote cholesterol synthesis [90]. Thyroid hormones also strongly induce the gene and protein expression of Apo A1 and scavenger receptor class B member 1 (SRB1), thereby increasing the cholesterol outflow of peripheral tissues into HDL in the cholesterol reverse transport pathway [91,92]. In addition, thyroid hormones can increase HDL metabolism by stimulating CETP activity [93]. In rats, thyroid hormones decrease serum levels and increase cholesterol clearance, mainly via the induction to hepatic LDLRs [92], which is also regulated by SREBP2 [94]. SREBP2 itself is reported to be transcriptionally regulated by thyroid hormones in rodents and humans [94]. Furthermore, thyroid hormone increased the transcription of LDLR-related protein 1 (LRP1), a lipoprotein associated with the clearance of chylomicron remnants and VLDL [95]. In liver, thyroid hormone also increased rat cholesterol 7α-hydroxylase (CYP7A1) expression, which plays a rate-limiting enzyme role in the process of converting cholesterol into bile acids in the reverse cholesterol transport pathway. CYP7A1 also reduce apolipoprotein B (the main apolipoprotein in LDL) expression to decrease serum LDL levels [96,97]. The excretion of bile acids in the liver and intestines is the last step of the reverse cholesterol transport pathway. Thyroid hormones promote this by stimulating ATP-binding cassette subfamily G member (Abcg5/Abcg8) complex gene transcription, whose effects are independent from LXRs [98].

## 5. Sex Hormones

It is widely observed that significant differences existed in plasma–lipid profiles between men and premenopausal women, the latter having a less proatherogenic plasma–lipid than men. Premenopausal women have higher HDL, LDL, VLDL, total plasma triglyceride, and VLDL triglyceride concentrations than age-matched men [99,100,101,102,103]. The difference in plasma–lipid concentrations likely accounts for at least part of the cardioprotective effect of female humans [104,105]. It is considered that the gender-based plasma–lipid differences were largely mediated by sex hormones; the fact that woman with polycystic ovary syndrome (PCOS) who develop hyperandrogenemia, and postmenopausal women appear, to undergo an increase in plasma triglyceride and LDL cholesterol and decrease HDL cholesterol concentrations [99,105,106,107,108] is one of the main pieces of supporting evidence.

(1)Estrogen decreases TG synthesis, while progesterone does the opposite.

The expression of transcription factor PPARy, as well as FAS and ACC1 mRNA levels in subcutaneous abdominal fat, was decreased in postmenopausal women treated with estrogen, these gene expressions being related to plasma triglyceride levels [109]. Aromatase knockout mice with estrogen deficiency increased the accumulation of hepatic triglycerides and the expression of FAS and ACC1 in liver, which could be reversed by estrogen replacement [110]. In human and rodent adipocytes cells, estrogen augments AMPK activation by increasing LKB expression, which favored local fatty acid oxidation and inhibition of lipogenesis [111].

Rats treated with progesterone have gained body weight and increased adipocytes, changes which may help to preserve the fat stores required for lactation in breast tissue [112]. Progesterone increased SREBP1 mRNA level in primary-cultured rat preadipocytes. In addition, membrane-bound and mature nuclear forms of SREBP1 protein accumulated in preadipocytes cultured with progesterone [113]. Although synthetic progestins generally increase LDL cholesterol and lower HDL cholesterol, it is not confirmed whether these changes result in increased incidence of cardiovascular disease [114].

(2)The role of testosterone.

Testosterone, in contrast to estrogen, appears to have a stimulatory effect upon DNL. Mice treated with dihydrotestosterone (DHT) exhibited increased body weight and visceral fat mass associated with triglyceride accumulation. DHT treatment increased expression of SREBP2 and FAS and decreased the inactivating phosphorylation of acetyl-CoA Carboxylase (ACC) [115]. Prenatal hyperandrogenism induces a decreased expression of Pgc1a and PPARα and overexpression of PPARy, Srebp, Chrebp, thus being more susceptible to hepatic steatosis and damage [116,117]. Consistent with this, DHT inhibits basal AMPK activation in a dose-dependent manner in human adipose cell lines by downregulating liver kinase B (LKB) expression [111]. On the other hand, animal studies suggested that testosterone deficiency enhances diet-induced liver fat accumulation [118]. One of the mechanisms by which testosterone regulates fat mass is stimulating lipid oxidation [119,120]. The body fat oxidation is enhanced by transdermal testosterone administration in hypogonadal men. The effect is also mediated by the modulation of other hormones, such as enhancing adrenergic action [121] and GH secretion [122], which in turn stimulates lipolysis and fat utilization. In high-fat-fed hepatic androgen receptor knockout male mice, hepatic steatosis will be developed through upregulation of genes involved in fatty acid (FA) synthesis and downregulation of genes involved in fat oxidation [123], indicating the importance of androgen receptor activation in determining hepatic lipid deposition. Similarly, administration of a 5a-reductase inhibitor in male obese Zucker rats induced hepatic steatosis [124]. Testosterone conversion to estradiol may play a role in the regulation of hepatic lipid metabolism. Aromatase-deficient men who cannot produce estradiol from testosterone will develop steatohepatitis [125]. In male mice with aromatase knockout, fatty liver occurs in parallel to hepatic FA uptake and an increase in de novo lipogenesis [126].

Woman after menopause and women with PCOS are prone to develop insulin resistance and obesity [108,127], which may be due to the changes in lipid homeostasis [128]. In fact, the conclusions of several studies [129,130,131] suggested that dyslipidemia associated with menopause was largely due to increasing age rather than decreased estrogen levels induced by ovarian function loss. It has been widely accepted that the cardiovascular protective lipid spectrum of premenopausal women is caused by sex hormones. However, this well-accepted view is not completely consistent with the results of some studies evaluating exogenous steroids effects on plasma–lipid dynamics and concentrations. Parenterally administered estrogens have either no effect or only very limited beneficial effects, whereas orally administered estrogens raised plasma triglyceride concentrations [132]. These results highlighted our currently inadequate understanding of the precise molecular mechanism by which sex hormones regulate lipid profiles, and how they coordinate different organ functions in the human body.

## 6. Growth Hormones

Growth hormone (GH) plays an important role in metabolic function throughout life cycle by regulating lipid and glucose metabolism. To date, more and more reports are suggesting that GH controls the liver’s substrate availability to regulate hepatocyte metabolism, which was largely due to the regulation of lipid mobilization in white adipose tissue (WAT) mediated by the GH receptor. Growth hormone is secreted intermittently by the pituitary gland with a large quantity at the beginning of slow wave sleep and less in a few hours after meals [133,134,135]. GH secretion is amplified during fasting and certain conditions such as type I diabetes [136,137,138], whereas a context of diet-induced obesity inhibits GH release [139,140,141]. Growth hormone secretion reaches maximum levels during puberty and was accompanied by very high circulating insulin-like growth factor (IGF-I) levels [142]. This tendency gradually decreased in adulthood [143].

GH is released into the general circulation and acts through the GHR to stimulate Igf1 gene expression, where the hepatocyte is the primary source of circulating IGF1 [144]. The actions of GH are mediated through the GHR which was expressed in a wide variety of tissues, and the signal transducer and activator of transcription 5 (STAT5) is the most studied regulatory system activated by GHR/Janus kinase 2 (JAK2) [145]. At the level of the hypothalamus, GH negatively feeds back through the GHR to reduce its own secretion [146]. In addition, IGF-I can suppress GH synthesis and release in pituitary cell through the IGF1R [147,148]. Specific knockout of the GHR or the IGF1R in pancreatic β-cell suggested that GH and IGF1 may augment β-cell proliferation and directly support optimal insulin synthesis and release [149,150,151]. Insulin, in turn, supports hepatic IGF1 production by stimulating IGF1 gene expression directly and maintaining hepatic GHR expression [152,153]. Additionally, insulin acts through INSRs on pituitary somatotropes to suppress GH secretion to counterbalance this system. As mentioned above, GH regulates IGF1 and insulin production and action, which in turn affects the production and function of GH itself.

(1)GH reduces de novo lipogenesis of adipose tissue.

There is evidence showing that GH reduces the DNL of adipose tissue [154], leading to significant fat mass loss. In the context of nutrient deprivation such as fasting and Type I diabetes, circulating insulin and IGF1 levels are reduced [155], while the fasting-induced rise in GH promotes white adipose tissue (WAT) lipolysis [156]. Additionally, the promotion of GH is not through a STAT5-dependent mechanism. Interestingly, accumulating evidence suggests that the constant activation of GH/IGF-I axis has a causal link with the consequent increase in insulin resistance [157]. GH decreases insulin sensitivity, thus preventing the substrate for lipogenesis from entering the cell [158]. For mice with specific knockout of STAT5 in adipocytes under chow-fed conditions, whose JAK-STAT signal pathway is dysregulated, increased adipose tissue mass will develop and improve body insulin sensitivity [159]. However, the loss of adipocyte STAT5 is neither associated with a reduction in WAT lipolysis nor does it impact the ability of GH treatment to reduce WAT mass [155]. These results suggested that WAT mass is affected by GH/GHR signaling in a STAT5-independent manner. In addition, consistent with this, the lipolytic effects were at least partly mediated via the hormone-sensitive lipase (HSL) in adipose tissue [160,161,162]. One of the underlying mechanisms may be enhancing agonist-induced stimulation of the β-adrenergic receptors (β-AR,) which participates in the activation of HSL [163]. Additionally, through GHR/JAK2-mediated ERK activation, GH inhibits adipogenesis and stimulates lipolysis in human adipocytes, consequently reducing PPARγ/FSP27 activity and stimulating HSL activity [164,165,166]. This lipolytic action mediated by growth hormones leads to an increase in circulating non-esterified fatty acids (NEFA) and glycerol which serve as substrates for hepatic gluconeogenesis, fatty acid (β) oxidation and ketogenesis. As the increased NEFA supply is positively associated with β-oxidation, it is not clear whether the GH has a direct stimulation of lipid β-oxidation [167]. Both in vivo and in vitro experiments failed to confirm that GH promotes β-oxidation [168,169]. However, β-oxidation is enhanced by hGH in human fibroblasts [167]. What is noteworthy is that GH may impact hepatic β-oxidation through STAT5-mediated suppression of BCL6, which was shown as a transcription factor to be a repressor of the β-oxidative program [170,171]. During adipocyte differentiation, GH can turn small adipocytes into larger and mature adipocytes, which are believed to have stronger lipolysis ability [172].

(2)New evidence for GH action on hepatic triglycerides.

Unlike fat cells, GH was considered to induce TG uptake and increase TG storage in liver [173,174]. Some mechanisms may be involved; the expression of hepatic HSL in bGH transgenic mice was significantly decreased, indicating that GH inhibited lipolysis of intrahepatic TG (IHTG) [174]. GH served to downregulate genes involved in lipid oxidation (e.g., PPAR-α, acyl CoA oxidase (ACO-1) carnithine palmitoyl transporter-1 (CPT-1)) and increased the expression of acetyl-CoA carboxylase (ACCβ) that promotes lipid synthesis in the liver [174,175,176]. However, increasing evidence is showing the association of low circulating GH levels [177] and IGF1 levels [178,179,180] with hepatic steatosis, and inactivating mutations in the GHR [181] of mice resulted in hepatic steatosis due to enhanced lipogenesis and reduced TG secretion from the liver [182]. For young men with abdominal obesity [183] and patients with primary GHD [184,185], GH treatment reduced their steatosis. The blockade of GH signaling using the GHR antagonist increased hepatic fat content in acromegalic patients [186,187]. Consistent with this, diet-induced obesity/steatosis mice exhibit reduced circulating GH levels [188,189]. Mouse models with specific knockout of hepatic GHR/JAK2/STAT5 signals exhibited a rise in circulating GH levels and reducing systemic insulin sensitivity, and thereby hyperinsulinemia, hyperglycemia, and WAT lipolysis [182,190,191]. These metabolic function changes shift the flux of glucose, glycerol, and NEFA to the liver, providing substrates for TG production. In these models, the fatty liver was thought to be due to the indirect action of GH. In the adult-onset loss of the hepatocyte GHR signaling model, the rapidly developed steatosis is associated with enhanced DNL and subsequently assessed in turn by the deuterated water labeling of newly formed fatty acids [192]. Thus, obesity-induced low GH levels may directly lead to the increase in DNL [193,194].

As the interplay between GH, IGF-1 and insulin is extremely complicated and, the mechanism of GH effect on lipid metabolism is not fully understand, more extensive studies are warranted in the future.

We summarized the effects of different hormones on lipid as shown in Table 1.

## 7. Adrenaline

As the response to emotional and physical stresses, adrenaline is released from the adrenal medulla, driving catabolic metabolism, increasing lipolysis and releasing NEFAs for oxidation [195]. Epinephrine binds to the membrane receptors of the G-protein-coupled receptor (GPCR) superfamily in target cells, and its effect on lipid metabolism is mediated through β-adrenergic receptors, in which β-3AR plays an indispensable role. Epinephrine binding leads to adenylyl cyclase activation, an increase in extracellular cAMP levels, and subsequent phosphorylation and activation of AMPK, thereby inhibiting adipogenesis. Adrenaline binds to the membrane receptors of the superfamily of G-protein-coupled receptors (GPCR) in target cells [196], and its impact on lipid metabolism is mediated via the β-adrenergic receptor, within which β-3AR plays an indispensable role in [197,198]. Epinephrine binding leads to the adenylyl cyclase activation, an increase cAMP levels inside cells, and subsequent phosphorylation and activation of AMPK, thereby inhibiting lipogenesis [9,197]. The observation of improved AMPK activation in WAT in the mice model with hypercyclic adrenaline further confirmed this view, and concomitantly increasing ACC phosphorylation was also observed. Additionally, in this model, ACC phosphorylation can be blocked by both a β-AR antagonist and AMPK inhibition which may contribute to reduce abdominal visceral fat accumulation and increase insulin sensitivity [199]. Many studies have shown the association between β3-AR gene polymorphism with insulin resistance, obesity and metabolic disorders such as coronary heart disease and hypertension [198,200,201]. Studies in Japan suggested the β-3AR gene polymorphism play a role in the genetic predisposition to increase small dense low-density lipoprotein (sdLDL) and shows small but significant effects on elevated LDL level, which closely associated with an increased risk of developing coronary artery disease [200,202]. Catecholamine resistance leads to lipid accumulation in adipose tissue by reducing lipolysis, increasing lipogenesis and impeding FFA transportation [203], which promote insulin signaling in adipose tissue [203]. A recent study demonstrated that epinephrine at physiological concentrations can induce extrapituitary prolactin (ePRL) expression in human monocytic cell line THP-1. Meanwhile, PRL has been shown to stimulate lipogenesis [204] which suggests the overall balance of adrenaline signal and ePRL signal could play a critical role in determining overall lipid turnover and accumulation in adipose depots [205].

## 8. Conclusions

Insulin plays a critical role in many hormones regulating lipid metabolism. The insulin-PI3K-AKT pathway and various downstream transcription factors were studied extensively. Insulin is right at the intersection of glucose and lipid metabolism, giving it an essential role in balancing the energy metabolism. However, more studies are needed to elucidate the regulation network. As FoxO6 does not undergo insulin-reliant phosphorylation and nuclear translocation like other FoxO subtypes. Instead, downregulation of FoxO6 suppresses expression of MTP, leading to amelioration of hypertriglyceridemia. In addition, the increase in extracellular cAMP induced by glucagon can enhance oxidation, which also provides a new idea for the treatment of fatty liver conditions. As TH acts on the mitochondrial genome, it endows TH with a special position in hormone regulation of lipid metabolism. A wide range of hormones regulate lipid metabolism simultaneously in a time-specific manner. In addition, hormone level varies over time, such as the GH, which increases significantly during adolescence, and can be compared in this sense to estrogen and progesterone in women of reproductive age. Regulation of lipid homeostasis may differ in fed and fasted states. Given the complexity of regulation network of in vivo situations, it is extremely difficult to assess the impact of individual hormones on animal models. However, the importance of understanding the hormonal regulation of lipid metabolism is obvious. Not only does it allow us to understand metabolic phenotypes characterized by hormone excess or deficiency, but also lead to the development of new targeted therapies that may modulate hormone. Studies about sex hormones on lipid metabolism suggested that sex hormone replacement can be used to treat dyslipidemia; however, clinical application must be strongly verified. Physiological concentration of adrenaline induces the expression of extrapituitary prolactin in adipose tissue macrophages, promoting fat weight loss. This observation also provides a novel weight control strategy. Given the clinical context of the global burden of obesity and its associated complications, morbidity and mortality, understanding and treating lipid accumulation in metabolic target tissues remains a priority.

## Figures and Tables

**Figure 1 molecules-27-07052-f001:**
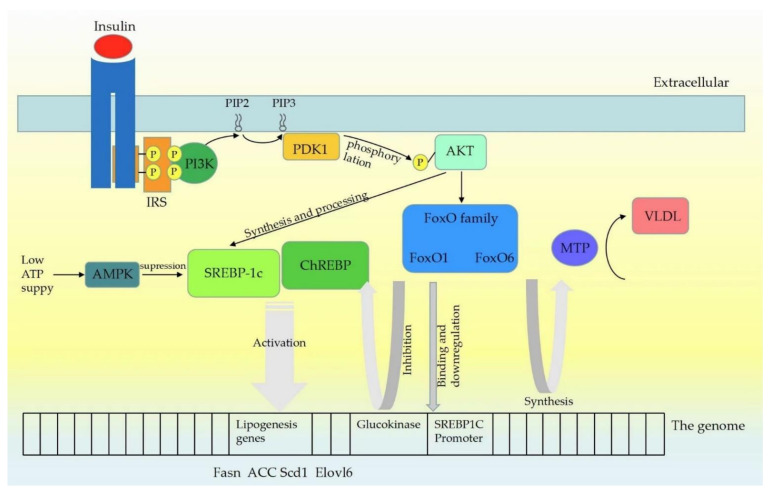
Insulin signaling pathway mechanism. Upon insulin receptors binding to ligands, leading to self-phosphorylation, insulin receptor substrates (IRS) are recruited and phosphorylated. IRS proteins are then recruited and activate PI3K, which phosphorylates PIP2 to generate PIP3. PDK1 is then activated by PIP3, which phosphorylates Akt at Thr308. Insulin activation of Akt enhances both the synthesis and processing of SREBP (SREBP1c mainly) in hepatocytes. SREBP1c and ChREBP activate lipogenic genes such as Fasn, ACC, Scd1 and Elovl6. At low cellular ATP levels, activation of AMPK interacts with and phosphorylates SREBP1c, thus inhibiting proteolytic cleavage and nuclear translocation, and repressing DNL. Similar to SREBP1c, FoxO proteins are also the class of transcription factors downstream of INS signaling pathway, and Akt controls the activity of the FoxO proteins through a phosphorylation mechanism. FoxO1 can bind to the SREBP1c promoter directly and further affect the transcriptional activity of Sp1 and SREBP1c. FoxO1 controls the expression of the glucokinase gene which may promote the lipogenic response by increasing glucose-6-phosphate levels and activating ChREBP. FoxO6 stimulated hepatic production of microsomal triglyceride transfer protein (MTP), which catalyzed the transfer of lipid to nascent apolipoprotein B (apoB), a rate-limiting step in the nascent assembly and secretion of VLDL.

**Table 1 molecules-27-07052-t001:** Summary of hormones and their effects on lipid production and decomposition.

Hormone	Lipid Synthesis	Lipid Catabolism
Insulin	Promotion SREBP-1C, ChREBP↑ [7,8,11] FoxO1↓ [15]	
Glucagon	Inhibition 1. SREBP-1C, ChREBP↓ [28,29] 2. ACC↓ [28,29]	Promotion 1. PPARα, Aox, CTP1, Ctp1a↑ [28,29] 2. ACC↓ [28,29] 3. HSL in adipocytes↑ [34,39,40,41]
Thyroid hormone	Promotion 1. SREBP-1C [56,57], ChREBP↑ [55] 2. ACC, Fasn↑ [65] 3. FATPs, L-FABPs↑ [60,61,62] 4. HMG-CoA Reductase↑ [75] Inhibition 5. PPARγ↓ [51]	Promotion 1. HSL [69,70,71], ATGL↑ [70,71] 2. PPARα↑ [53] 3. Lipophagy↑ [80] 4. PGC1α, mtTFA↑ [82] 5. MCAD [86], PDK4 [87], UCP2↑ [88] 6. CYP7A1↑ [76]
Estrogen	Inhibition 1. PPARγ↓ [110] 2. ACC, Fasn↓ [110]	
Progesterone	Promotion 1. SREBP↑ [113]	
Androgen	Promotion 1. SREBP, ChREBP↑ [116,117] 2. ACC, Fasn↑ [115]	Promotion [119,120]
Growth homone	Inhibition 1. PPARγ↓ [164,165,166] 2. Reducing insulin sensitivity [158]	Promotion 1. WAT lipolysis↑ [156] 2. HSL↑ [160,161,162]

SREBP, ChREBP and FoxO are important transcription factor regulating lipid production. PPARα promotes FFA β-oxidation by stimulating the transcription of FFA β-oxidation genes such as Aox, Ctp1a. PPARγ plays a role in promoting lipid synthesis. HSL and ATGL are enzymes that catalyze lipolysis. MCAD, PDK4, UCP2 are important enzymes in β-oxidation. FATPs and L-FABPs are protein transporters which mediate FFA-entering hepatocytes for substrates of liver TG synthesis.

## Data Availability

The data presented in this study are available on request from the corresponding author.

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
