# Peer review of "Important Hormones Regulating Lipid Metabolism"

_molecules, 2022, doi:10.3390/molecules27207052_

Round 1
Author Response
Reviewer1
This manuscript entitled “Important hormones regulating lipid metabolism” is a very useful study in which focused on lipid metabolism to encounter the major problem of obesity. The obesity is a big issue gaining importance in present time after a COVID-19 pandemic. This issue has to be well understood at molecular level (hormonal level) to find an effective solution. The studies like these will boost the current research to achieve the goal of minimizing the problem of obesity. The present manuscript is nicely written but there are certain mistakes included in comments below need to be addressed before going further.
Comment 1: In the abstract section “In this review, we will summarize the current landscape” in this sentence “we will” has been used it must me “we have” for correct framing of sentence.
Response: We have replaced “we will summarize” by “we have summarized”.
Comment 2: In the abstract section “We will comprehensively evaluate the real situation” “We will” should be replaced with “We have”.
Response: In the abstract section, “We will” is replaced by “We have” according to the comment.
Comment 3: In Figure 1 there is good pathway which has been explained in details in its legend. But in figure itself some miner additions can be done to make clear the basic theme of figure. Additions like PDK1 phosphorylate Akt the word “phosphorylation” can be added in between PDK1 and Akt on arrow. Similarly, as Akt enhances both the synthesis and processing of SREBP on arrow between Akt and SREBP the “synthesis and processing” word can be added. SREBP1c and ChREBP is able to activate lipogenic genes like “Fasn, Scd1 and Elovl6” but in figure 1 “Fas, ACC, Scd1, Elovl6” are labelled make this clear keep same and also add the word “activation” near to theses genes. Once again see how this image can be reframe to make clear presentation of the pathway for simple and easy understanding of reader.
Response: In figure1, we have made some changes to make it more clear, such as adding word like “phosphorylation” between PDK1 and Akt on arrow, “synthesis and processing” was added on arrow between Akt and SREBP as requested. And we have checked the label and modified the figure to keep it consistent with “Fasn, Scd1 and Elovl6” in the content of the text. Besides, we made efforts to embed more information to make this figure more beautiful and easier to understand.
Comment 4: In “Glucagon” section “very-low-density lipoprotein (VLDL)” this abbreviation has used and in previous section only “VLDL” is used do proper full form at initial text then use only abbreviation in later part of the text. Avoid repeated use.
Response: We have checked the use of abbreviations to make sure the proper full form at first and followed abbreviations at latter text such as triglycerides (TG), Very low-density lipoprotein (VLDL), Adipose triglyceride lipase (ATGL), Pyruvate Dehydrogenase Kinase 1 (PDK1), sterol regulatory element binding protein 1c (SREBP1c), etc.
Comment 5: The points like “Glucagon” Thyroid hormone” “Sex hormones” Growth hormones” “Adrenaline” is not having any diagrammatic representation. The readers may feel difficult to understand the role and pathways of each such hormone without figure. If possible add figures in each part of these hormones for better understanding of reader. However, these points are well written.
Response: In the points like “Glucagon” Thyroid hormone” “Sex hormones” Growth hormones” “Adrenaline”, figures and subsequent subtitle were added for better understanding. In the Glucagon part, “(1) At the molecular level” and “(2) At the level of organ and in vivo” were marked at the beginning of the paragraph to describe the role of glucagon in a hierarchical manner and in organized order. Similarly, in the Thyroid hormone part, “(1) The role of thyroid hormone receptors regulating lipid metabolism.” “(2) TH increases hepatic TG synthesis” “(3) TH reduces the content of TG in hepatocytes and adipocytes by promoting TG catabolism.” “(4) Mitochondria, an important target for TH regulating lipid metabolism” “(5) TH reduces serum cholesterol levels” were added. In the Sex hormones part, “(1) Estrogen decreases TG synthesis, while progesterone does the opposite” “(2) The role of testosterones” were added. In the part of Growth hormones”(1) GH reduces the de novo lipogenesis of adipose tissue.” “(2). New evidence for GH action on hepatic triglycerides” were added. All in a word, figures and subtitles were added for better describe the effect and the role of each hormone in a clearer order.
Comment 6: The conclusion section encompasses the need of understanding lipid metabolism in relation to its controlling hormones. This section can be added with some future targets important hormones to control obesity. There should be clear and conclusive statements about what needs to be done to control the problems like obesity and the future guidelines to encounter this problem to make conclusion effective.
Response: In the conclusion, as shown after modification, we summarized the whole review, for the emerging pathways that play an important role in regulating lipid metabolism, we assumed some possibilities about new corresponding targets to improve lipid metabolism at the part.
Comment 7: There is no any tabular representation in this review. If possible add one table in which there is information about hormone and its role in lipid metabolism/catabolism for better understanding of reader.
Response: A table summarizing the role of various hormones in lipid synthesis and catabolism was updated.
Comment 8: The manuscript is nicely written but there are few places where text formatting is not proper. The manuscript has to be checked for rectification of grammatical and punctuational mistakes before taken into further consideration.
Response: We have checked and improved the whole manuscript to avoid text formatting, grammatical and punctuational mistakes.
Comment 9: The manuscript has the potential to be considered for publication but all the mistakes and errors with scientific comments has to be rectified in major revision.
Response: For the major revision, we checked the logic of the full text to avoid scientific errors.
Reviewer 2 Report
The reviewed manuscript summarizes the current information about hormones involved in regulation of lipid metabolism.
As an idea it is perhaps justifiable, however, manuscript presents several important issues and drawbacks.
1) Since lipids and their metabolism is so important as stated in the abstract, some introductory paragraphs concerning lipid status and regulation are clearly missing.
2) Informaiton about individual hormones is very detailed but clmsily arranged and expressed which make it very difficult for a reader to find an coherence.
Overall impression is very ambiguous. The entire manuscript seems to be unbalanced and difficult to read.
Language must be improved.
Abtract must be completely revised. In the current state it is not acceptable.
Author Response
Reviewer:2
The reviewed manuscript summarizes the current information about hormones involved in regulation of lipid metabolism.
As an idea it is perhaps justifiable, however, manuscript presents several important issues and drawbacks.
Comment 1: Since lipids and their metabolism is so important as stated in the abstract, some introductory paragraphs concerning lipid status and regulation are clearly missing.
Response: A introductory paragraph concerning lipid status and regulation was added between the abstract and the “insulin” part. This paragraph summarized the whole lipid family, important organs and enzymes concerned in the process of lipid metabolism.
Comment 2: Informaiton about individual hormones is very detailed but clmsily arranged and expressed which make it very difficult for a reader to find an coherence.
Response: We have made the following modifications to address this issue: 1. Figure 1 was re-made with more clearer signaling pathways symbols and more essential textual content, making it more understandable. 2. We have re-organized the sentences especially in the parts of “Glucagon” “Thyroid hormone” “Sex hormones”, improving logic for better understanding. 3. The number and subsequent subtitle were added for better describe the effect and the role of each hormone in a clearer order, which also highlighted the key point of each paragraph.
Comment 3: Overall impression is very ambiguous. The entire manuscript seems to be unbalanced and difficult to read. Language must be improved. Abstract must be completely revised. In the current state it is not acceptable
Response: We have revised the abstract to minimize any grammatical mistake or logical errors.
Round 2
Reviewer 1 Report
Manuscript Title: Important hormones regulating lipid metabolism.
Manuscript ID: Molecules-1939736.
Revision-1
This manuscript entitled “Important hormones regulating lipid metabolism” after first revision sounds good in terms of compiling valuable information. But still, there are certain mistakes and corrections that need to be done before going further with the decision regarding this manuscript.
Comment 1: In the legend of Figure 1 “At low cellular ATP levels, activation of ATMP interact with and phosphorylate SREBP-1c” This is the explanation. But in a figure, the “at low ATP level AMPK seen to be interacting with phosphorylate SREBP-1c” Again see the figure carefully and do all the necessary corrections to make an exact figure with a clear understanding.
Comment 2: In an earlier manuscript before revision Figure 1 lipogenic genes such as “Fas, ACC Scd1, and Elovl6” were mentioned but after revision in Figure 1 “Fasn, Scd1, and Elovl6” are mentioned make it clear and mention properly which are the actual lipogenic genes. While drawing the figure carefully label and mention all the minute details for an exact understanding of the pathway.
Comment 3: “Table 1: Summary of hormones and their effects on lipid production and decomposition.” added after revision that contains very important information but if possible add at least one reference for each hormone and their activities. The text in the table must possess proper referencing to support all the statements and activities mentioned regarding each hormone.
Comment 4: In the conclusion section “However, more studies are needed to elucidate the regulation network” the author wants to say “However, more studies are needed to elucidate the glucose regulation network”. However, the present form of the conclusion is nicely written and it is with meaningful sentences with valuable information. Just recheck again and add any missing words to fulfill the right meaning of all the sentences.
Comment 5: Proofread the whole manuscript for grammar and make corrections if needed. However, the quality of information is now satisfactorily improved and the manuscript can be considered positively after these minor corrections.
However, the manuscript of present forms has been improved a lot and contains valuable information but after doing minor revisions as per the suggestions given the manuscript can be considered positively for publication.

Author Response
Response to reviewer 1
Thank you very much for the comments, we have followed your suggestions to revise the manuscript.
- 1. In the legend of Figure 1 “At low cellular ATP levels, activation of ATMP interact with and phosphorylate SREBP-1c” This is the explanation. But in a figure, the “at low ATP level AMPK seen to be interacting with phosphorylate SREBP-1c”, We read the original literature again, in figure, the “AMPK” is correct, and the "ATMP" in the explanation is atypo, we have replaced it with "AMPK".
- In the figure, “Fas, ACC, Scd1, and Elovl6” are the actual and correct lipogenic genes, so are the explanation. We have made correctionin the figure and explanations in the revised manuscript.
- 3. In table1, in therevisedmanuscript, we have followed your comment to add references for each hormone and their activities to support the statements.
- 4. We double-checked the conclusion part, “However, more studies are needed to elucidate the regulation network” was replaced with “However, more studies are needed to elucidate the glucose and lipid regulation network” to express proper meaning.
- 5. We proofread again the whole manuscript to minimize grammatical errors.
Reviewer 2 Report
Authors may have made the suggested changes but these are not easily follwed in the present mode of text changes. Please, provide your revised manuscript where all changes are accepted and resubmit.
Author Response
Response to reviewer 2
Thank you very much for the comments, we have resubmitted the revised manuscript where all changes are accepted.